

# Assessing tooth wear progression in non-human primates: a longitudinal study using intraoral scanning technology

Ian Towle[1], Kristin L. Krueger[2], Raquel Hernando[1,3] and Leslea J. Hlusko[1]

[1] Centro Nacional de Investigación sobre la Evolución Humana (CENIEH), Burgos, Spain
[2] Department of Anthropology, Loyola University Chicago, Chicago, IL, United States of America
[3] Institut Català de Paleoecologia Humana i Evolució Social (IPHES), Tarragona, Spain

## ABSTRACT

Intraoral scanners are widely used in a clinical setting for orthodontic treatments and tooth restorations, and are also useful for assessing dental wear and pathology progression. In this study, we assess the utility of using an intraoral scanner and associated software for quantifying dental tissue loss in non-human primates. An upper and lower second molar for 31 captive hamadryas baboons (*Papio hamadryas*) were assessed for dental tissue loss progression, giving a total sample of 62 teeth. The animals are part of the Southwest National Primate Research Center and were all fed the same monkey-chow diet over their lifetimes. Two molds of each dentition were taken at either two- or three-year intervals, and the associated casts scanned using an intraoral scanner (Medit i700). Tissue loss was calculated in *WearCompare* by superimposition of the two scans followed by subtraction analysis. Four individuals had dental caries, and were assessed separately. The results demonstrate the reliability of these techniques in capturing tissue loss data, evidenced by the alignment consistency between scans, lack of erroneous tissue gain between scans, and uniformity of tissue loss patterns among individuals (*e.g.*, functional cusps showing the highest degree of wear). The average loss per $mm^2$ per year for all samples combined was 0.05 $mm^3$ (0.04 $mm^3$ for females and 0.08 $mm^3$ for males). There was no significant difference in wear progression between upper and lower molars. Substantial variation in the amount of tissue loss among individuals was found, despite their uniform diet. These findings foster multiple avenues for future research, including the exploration of wear progression across dental crowns and arcades, correlation between different types of tissue loss (*e.g.*, attrition, erosion, fractures, caries), interplay between tissue loss and microwear/topographic analysis, and the genetic underpinnings of tissue loss variation.

## INTRODUCTION

Assessment of dental tissue loss, including through wear, fracture and pathologies, provides insights into diet, behavior, functional adaptations, and broader ecological contexts of extant and extinct primates (*e.g.*, *Crovella & Ardito, 1994*; *Hillson, 2001*; *Kaifu et al., 2003*; *de Castro et al., 2003*; *Ungar & Sponheimer, 2011*; *Meng et al., 2011*; *Benazzi et al., 2013*;

Corresponding author
Ian Towle, ianetowle@hotmail.co.uk

*Towle et al., 2022*; *Towle et al., 2024*). Typically, wear and pathology studies either focus on specific instances (*e.g.*, pathological case studies) or adopt a broader perspective by considering population averages, often relying on the frequency patterns of tooth wear and fracture scores (*e.g.*, *Eshed, Gopher & Hershkovitz, 2006*; *Esclassan et al., 2009*; *Shykoluk & Lovell, 2010*; *Lanigan & Bartlett, 2013*; *Fannin et al., 2020*; *Silvester, 2021*; *Towle et al., 2021a*). While these approaches are crucial for understanding diet and behavior, they lack the ability to contextualize how attrition, abrasion, erosion, caries, or fractures dynamically interplay with each other over time, but also with the underlying tooth morphology and masticatory cycle. With advancements in clinical dentistry technology it is now feasible to scan large numbers of dentitions to reliably collect tissue loss information through time. In this study we explore the potential of applying these techniques, designed explicitly for use in a clinical human setting, to other primates.

Intraoral scanners are gaining prominence as an essential tool in clinical dentistry, revolutionizing orthodontic treatments and dental restorations (*Revilla-Leon et al., 2021*; *Valenti & Schmitz, 2021*; *Huqh et al., 2022*; *Akl, Mansour & Zheng, 2023*). These scanners have been shown to produce consistent results among devices, and although the scanner with the best accuracy/precision varies among studies, these values are typically high in all brands/models tested (*e.g.*, *Abduo & Elseyoufi, 2018*; *Amornvit, Rokaya & Sanohkan, 2021*; *Afrashtehfar, Alnakeb & Assery, 2022*; *Borbola et al., 2023*; *Selvaraj et al., 2023*). Their potential has also been demonstrated to accurately measure dental tissue loss, which holds great utility in a clinical setting. This is particularly beneficial for tracking pathology and wear progression through time (*e.g.*, *O'Toole et al., 2019a*; *García et al., 2022*; *Marro et al., 2022*; *Schlenz et al., 2022*; *Mitrirattanakul et al., 2023*). Specific software designed to measure dental tissue loss using intraoral scanner 3D models has been developed (*e.g.*, *WearCompare*; *O'Toole et al., 2019a*) and has demonstrated reliability even over short time intervals (*O'Toole et al., 2019a*; *O'Toole et al., 2020*; *O'Toole et al., 2023*). For this type of measurement, intraoral scans are taken at separate time points and then superimposed to quantify differences (*e.g.*, total volume loss or volume loss per $mm^2$ of surface analyzed).

Our study investigates the potential use of an intraoral scanner and associated software to quantify dental tissue loss in a captive hamadryas baboon (*Papio hamadryas*) sample. The selection of this sample allows for a focused examination of dental tissue loss within a controlled primate group (*i.e.,* with a uniform, known diet) enabling insights into wear progression of non-human primates. Additionally, the study seeks to assess whether the same equipment and techniques employed in a clinical human setting can be successfully utilized in non-human primates. By creating 3D models of upper and lower second molars, the investigation quantifies the extent of tissue loss over two distinct time periods—two years and three years. Similar time intervals have been used in a clinical human setting, allowing direct comparisons in terms of scan alignment and tissue loss. The study's objectives are twofold: first, to determine the effectiveness of intraoral scanning technology in accurately capturing dental tissue loss in non-human primates; and second, to shed light on the patterns and magnitudes of tissue loss, variability among individuals, and limitations in terms of scan alignment. We hypothesize that due to higher levels of tooth

wear in baboons compared to people alive today, who generally consume soft, heavily-processed, diets, the scanner-based method of quantifying tissue loss may be less reliable. This is because, all else being equal, it should become more difficult to align scans with increasing wear progression.

## MATERIAL AND METHODS

Portions of this text were previously published as part of a preprint (*Towle et al., 2024*). This study utilized a sample of captive pedigreed baboons that were part of a larger breeding colony at the Southwest National Primate Research Center in San Antonio, Texas. The sample consists predominantly of olive baboons (*Papio hamadryas anubis*), yellow baboons (*P. hamadryas cynocephalus*), and their hybrids (all classified in the present study as within a single species: *P. hamadryas*, following *Jolly (1993)*, to maintain consistency with prior research on this colony, *e.g.*, *Hlusko, Sage & Mahaney, 2011*; *Hlusko et al., 2016*). The colony was maintained in pedigrees (*i.e.,* all mating controlled), with a female to male sex ratio of approximating 2:1 (*Hlusko et al., 2004*; *Hlusko, Lease & Mahaney, 2006*). Baboons have a rich evolutionary history being explored and revealed through paleontological and molecular data (*e.g.*, *Brasil et al., 2023*; *Kopp et al., 2023*; *Sørensen et al., 2023*). Our study focuses on the variation of dental tissue loss just within this one captive population.

As part of previous studies by one of the authors (LJH) and collaborators, dental molds of these baboons were taken in the late 1990s and early 2000s (*Hlusko & Mahaney, 2003*; *Hlusko et al., 2004*; *Hlusko, Lease & Mahaney, 2006*). Detailed methodology can be found in these earlier works, with the information important for the present study summarized here. Whilst individuals were anesthetized, both the maxillary and mandibular postcanine dentitions were molded using a rapidly setting high-resolution impression material commonly used by dentists (Coltene President Plus Jet regular body silicon). Positive casts were poured with dental plaster within one week of molding. Out of the more than six hundred baboons for which casts were created, a small subsample had their dentitions molded on more than one occasion. Within this subsample, 31 individuals had their dentitions molded with two years (eight individuals) or three years (23 individuals) between moldings (± one month, with one exception with a couple of weeks more; Table 1). All individuals had a uniform diet of monkey chow, and could eat and drink water *ad libitum* (*Hlusko & Mahaney, 2003*; *Willmore et al., 2009*).

Four individuals in the three-year interval group had a carious lesion in either the upper or lower molar, and were excluded from the primary analysis. The remaining 27 individuals had pathology-free second molars and displayed moderate degrees of wear at the time of the first impression (defined as no coalescence of dentine islands on the occlusal surface). The complete dataset is available in the SOM, which also provides further information for each individual studied, including molding dates, date of birth and death, specimen numbers and the tooth studied (*i.e.,* left or right second molars). Each individual is associated with two specimen numbers: the original SNPRC number, and the current specimen number for the osteological specimens housed at Loyola University Chicago (W number). While both numbers are provided in the SOM, we use just the current W number in the main

**Table 1 Sample information, including specimen number, sex, caries presence, and age at first and second mold and death (in years).** The exact dates of birth, moldings and death are presented in the Supplemental Information.

| Specimen (W #) | Sex | Caries | Age at first mold | Age at second mold | Age at death |
|---|---|---|---|---|---|
| 880 | Female | No | 12.7 | 15.7 | 24.4 |
| 957 | Female | No | 13.1 | 16.2 | 23.6 |
| 420 | Female | No | 21.2 | 24.3 | 27.3 |
| 515 | Female | No | 17.7 | 20.7 | 25.2 |
| 296 | Female | No | 19.3 | 22.4 | 24.1 |
| 317 | Female | No | 18.4 | 21.5 | 23.2 |
| 312 | Female | No | 11.5 | 14.7 | 16.4 |
| 469 | Male | No | 10.9 | 14.0 | 17.8 |
| 362 | Male | No | 12.1 | 15.2 | 17.9 |
| 847 | Female | No | 11.2 | 14.4 | 21.7 |
| 731 | Male | No | 11.1 | 14.3 | 20.8 |
| 864 | Male | No | 11.1 | 14.3 | 20.8 |
| 656 | Female | No | 13.8 | 16.8 | 22.7 |
| 398 | Female | No | 11.9 | 15.0 | 18.4 |
| 231 | Female | No | 11.4 | 14.4 | 14.6 |
| 826 | Male | No | 10.4 | 13.4 | 20.1 |
| 619 | Female | No | 19.5 | 22.7 | 27.5 |
| 443 | Female | No | 10.2 | 13.4 | 15.0 |
| 955 | Female | No | 9.6 | 12.7 | 20.0 |
| 529 | Female | No | 22.4 | 24.3 | 25.0 |
| 525 | Female | No | 22.5 | 24.4 | 25.2 |
| 228 | Female | No | 31.5 | 33.4 | 33.7 |
| 437 | Female | No | 23.7 | 25.5 | 27.2 |
| 393 | Female | No | 19.2 | 21.3 | 24.2 |
| 236 | Female | No | 19.0 | 21.0 | 21.0 |
| 284 | Female | No | 18.6 | 20.4 | 21.9 |
| 583 | Female | No | 21.1 | 23.1 | 28.7 |
| 279 | Male | Yes | 16.7 | 19.9 | 21.7 |
| 246 | Female | Yes | 14.4 | 17.5 | 18.0 |
| 654 | Male | Yes | 12.1 | 15.1 | 20.9 |
| 288 | Female | Yes | 9.6 | 12.7 | 14.7 |

text, tables and figures for simplicity and to be consistent with how CT scans of these same specimens are labeled in MorphoSource.

Casts were scanned (by IT) using an intraoral scanner (Medit i700; manufacturer specified accuracy: 10.9 µm ± 0.98), with the high-resolution setting selected to produce an STL file for each scan. In addition to the two scans of each individual (from the baseline and secondary casts), for three specimens the osteological specimen was also scanned (by KLK), adding a third time point to assess specific types of wear or pathology over time. These included one specimen with a carious lesion and two specimens with tooth fractures. All STL files were exported from the Medit Link software to *Meshlab* (*Cignoni et al., 2008*)

where the files were trimmed to only include the posterior teeth of the side under study (left or right). The same side tooth was used for both upper and lower within each individual, with preference given to the side with fewer artifacts such as saliva or food debris, or potential molding or casting flaws. Small artefacts were removed through cutting and hole filling in Meshlab. The entire original 3D model in STL format (*i.e.,* not just the cropped region of interest for this study) for each cast and osteological specimen is available open access on Zenodo (https://doi.org/10.5281/zenodo.11801183).

A purpose-built freeware, *WearCompare* (Leeds, UK: https://leedsdigitaldentistry.com/wearcompare/), was used to estimate the tissue loss data, following the protocols set out in *O'Toole et al. (2019a)* and *O'Toole et al. (2020)*. This involved loading the baseline and secondary scans (taken either two or three years apart) followed by superimposition. This process involves aligning the surfaces using the global recognition feature, which uses an iterative closest point algorithm. Subsequently, a selective surface alignment protocol was used, focusing on further alignment of the particular tooth under study. In this protocol, buccal and lingual surfaces were selected as reference areas, and a refined iterative closest point alignment performed (*O'Toole et al., 2020*).

The occlusal surface was selected as the measurement surface following the methodology outlined by *O'Toole et al. (2019a)*. The main variables of interest were volume loss over the entire molar occlusal surface ($mm^3$) and the volume loss per $mm^2$ of surface, following *O'Toole et al. (2020)*. Additionally, the maximum surface deviation, which is the point at which the most tissue has been removed (measured vertically from the occlusal surface in microns) was recorded. Alignment quality was assessed based on comparisons with *O'Toole et al. (2020)*, in which poor alignment was defined as less than 75% of data points coinciding within 25 $\mu$m of each other. All *WearCompare* analyses were performed by a single operator (IT). Descriptive statistics were calculated in Microsoft Excel to summarize tissue loss data. Statistical analyses were conducted to investigate differences in dental tissue loss (volume loss per $mm^2$), between individuals with two years *vs.* three years of wear (females only), males and females (three-year interval individuals only) and between upper and lower molars across all individuals (males and females). Based on assessments of normality and homogeneity, a two-sample *t*-test was used for the first two comparisons, while a Mann–Whitney U test was used for the upper *vs.* lower comparisons (alpha level, $p < 0.05$). All statistical analyses were performed in R (version 4.2.2; *R Core Team, 2019*).

Box and whisker plots were created to compare dental tissue loss metrics across different subsets of the baboon sample. Scatter plots and linear regression analyses were performed to investigate the relationship between scan alignment and two different dental measurements: volume loss per $mm^2$ and occlusal area ($mm^2$) for all second molars combined (excluding the four individuals with dental caries). Regression lines were fitted to assess the overall trend across all specimens, as well as separate trends split by sex. For the creation of the figures (box and whisker; scatter plots with trend lines), data were processed and visualized using Python (version 3.8) in a Jupyter notebook environment, and using the pandas (version 1.2.3), seaborn (version 0.11.1) and matplotlib (version 3.3.4) libraries. Summary statistics for each regression analysis were performed in R (version 4.2.2; using the "car" package).

**Table 2** **Tissue loss values split into time intervals between molds and sex (all males had a three-year interval).** Scan alignment information is also provided and is the percentage (%) of data points that coincide within 25 μm of the two scans. Note: the four individuals with dental caries are not included.

| Variable | Female: Two years ($N = 8$) | Females: Three years ($N = 14$) | Males: Three years ($N = 5$) | All samples ($N = 27$) |
|---|---|---|---|---|
| **Upper second molar** | | | | |
| Total volume lost ($mm^3$) | 5.98 | 8.74 | 19.70 | 9.95 |
| Volume loss per $mm^2$ ($mm^3$) | 0.07 | 0.13 | 0.25 | 0.13 |
| Maximum surface deviation (microns) | 476.39 | 651.53 | 939.48 | 652.96 |
| Scan alignment (%) | 93.58 | 89.32 | 89.28 | 90.57 |
| **Lower second molar** | | | | |
| Total volume lost ($mm^3$) | 6.05 | 9.99 | 19.98 | 10.67 |
| Volume loss per $mm^2$ ($mm^3$) | 0.08 | 0.13 | 0.23 | 0.14 |
| Maximum surface deviation (microns) | 540.60 | 655.63 | 1,044.54 | 693.57 |
| Scan alignment (%) | 93.73 | 89.14 | 88.58 | 90.40 |

## RESULTS

The values for the three tissue loss variables split by sex and years between molds, for non-carious specimens, are presented in Table 2. On average, females exhibited a tissue loss of 3.08 $mm^3$ per year (with an average of 3.01 $mm^3$ per year for the two-year interval group and 3.12 $mm^3$ per year for the three-year interval group), while males had an average tissue loss of 6.61 $mm^3$ per year (all males were in the three-year interval group). The average loss per $mm^2$ per year for all samples combined was 0.05 $mm^3$ (0.04 $mm^3$ for females and 0.08 $mm^3$ for males). Notably, there was substantial variation among individuals within each group category (*e.g.*, male/female, two-years interval/three-year interval; see Fig. 1). The average maximum surface deviation for the two-year interval group, among females only, was 508.50 μm, while for the three-year group it was 653.58 μm. These findings collectively suggest an increase in tissue loss with longer periods of time, as expected.

Much of the observed tissue loss is associated with functional cusps, meaning the buccal side in lower molars and lingual in upper molars (refer to Fig. 2). The maximum surface deviation consistently occurs on the functional cusps of the crown in all but one of the non-carious teeth studied (*i.e.,* one out of 54 teeth; this outlier is discussed in more detail below). There is minimal variation between upper and lower molars, with no significant differences between the dentitions for volume loss per $mm^2$ (Mann–Whitney $U = 368.5$, $n = 27$, $p = 0.952$). Among females, those with a three-year interval between molds exhibited a significantly greater amount of tissue loss than those with two-year intervals between molds, as indicated by volume loss per $mm^2$ ($t = 3.22$, n1 = 28, n2 = 16, $p = 0.002$). Furthermore, males showed significantly more wear than females, with volume loss per $mm^2$ serving as the appropriate proxy to account for differences in tooth size (three-year time-interval for both; t = $-4.03$, n1 = 28, n2 = 10, $p < 0.001$).

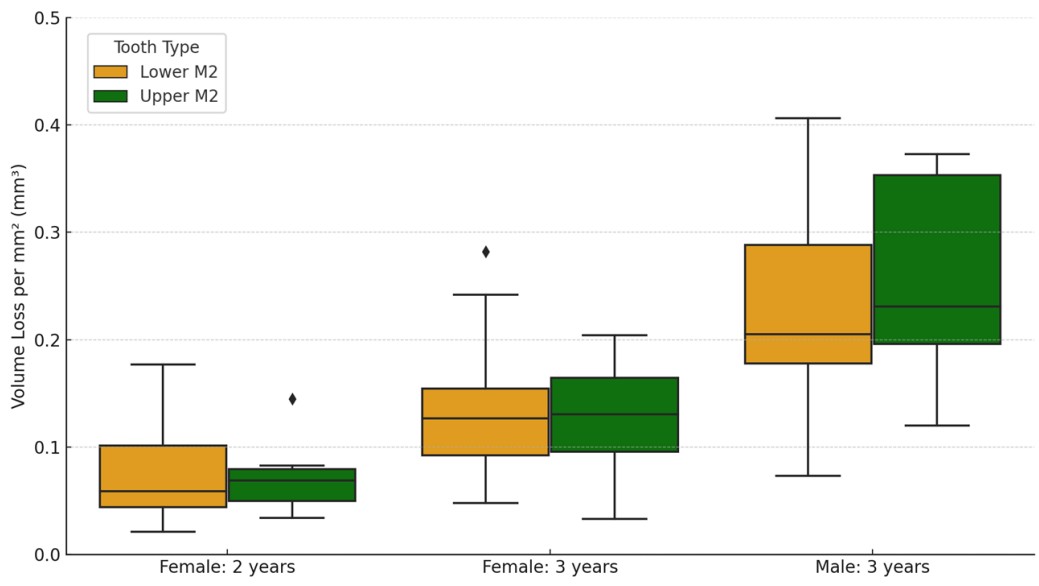

**Figure 1** **Box plots showing the variation in values for both upper and lower second molars (M2), split by females with two-year interval between molds, females with three-year interval between molds, and males with three years interval between molds.** There were no males with a two-year interval between molds.

The four teeth with occlusal caries exhibit higher tissue loss compared to their non-carious counterparts (Table 3; Fig. 2). The opposing upper molar in two of the individuals with caries in the lower molar show a reduction in tissue loss compared to non-carious teeth (compared to the other teeth with a three-year interval between molds for their sex). However, in the other two individuals, the opposing tooth shows high tissue loss values, especially in one specimen, W 288. In this case, although a deep carious lesion is forming in the occlusal basin of the upper right second molar, it is the lower right second molar that shows the most extreme tissue loss, resembling erosion or attrition, although a severe form of caries cannot be ruled out (see Fig. 3). The upper second molar of specimen W 654 shows the most tissue loss among all samples studied (over 100 mm³), with the deepest point (*i.e.,* area with most tissue removed between molds) reaching almost 6 millimeters. This extensive tissue loss is attributed to caries, which has removed substantial parts of the tooth crown over the three years between the molds.

Scan alignment was above the threshold used in *O'Toole et al. (2020)* for all samples, in which poor alignment was defined as less than 75% of data points coinciding within 25 $\mu$m of each other. Scan alignment decreased from the two-year interval group to the three year-interval group (Table 2), and a similar trend is observed when comparing volume loss per mm$^2$ with alignment percentage (Fig. 4). In particular, the linear regression trend depicted in Fig. 4 indicates that as the volume loss per mm$^2$ increased, there is a corresponding decrease in scan alignment. This trend remains when split into male and females, although both show only a weak negative relationship, with neither reaching statistical significance (Table 4). Similarly, scan alignment and occlusal area showed a very

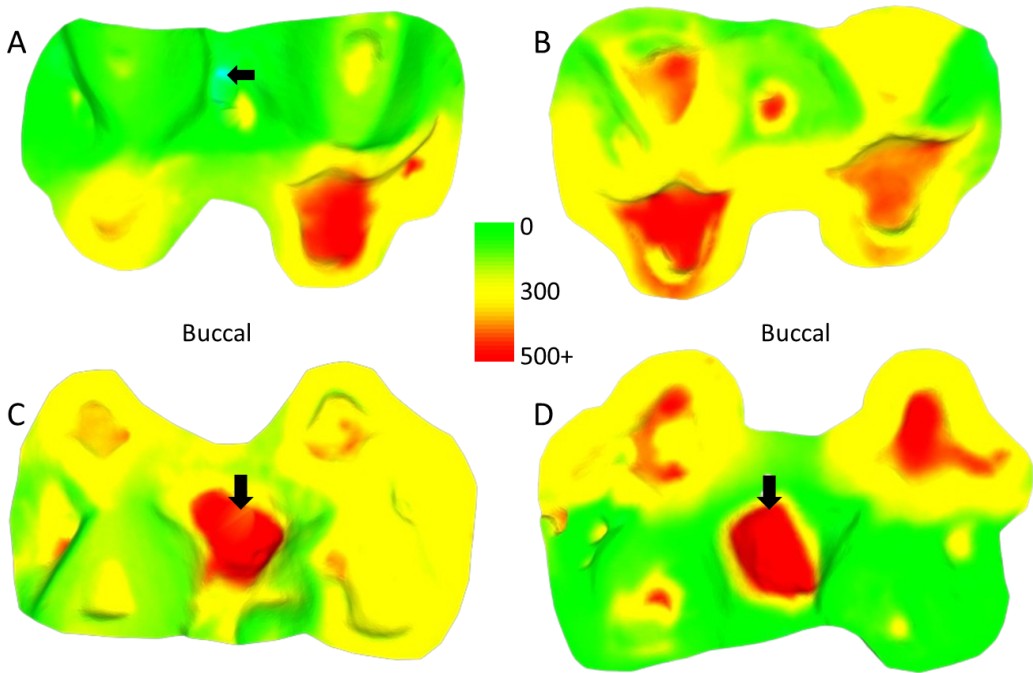

**Figure 2** **Comparison of dental tissue loss in lower second molars in occlusal view.** (A) Specimen W 236 (lower right second molar): two years of attrition; (B) specimen W 420 (lower right second molar): three years of attrition. Both (A) and (B) show greater tissue loss on the buccal cusps; (C) lower left second molar of W 279; (D) lower right second molar of W 246. Both (C) and (D) show large carious lesions that developed significantly during the three years between the first and second molds in both cases (black arrows indicate the carious lesions). The scale bar is in microns, and represents the distance difference between the first and second molds. The black arrow in (A) is a negative value from the first mold to the second mold, likely a small artifact, caused by saliva or food debris in the second mold. Figure created using WearCompare (https://leedsdigitaldentistry.com/wearcompare/).

**Table 3** **Tissue loss in individuals with dental caries (bold indicates the tooth affected).** All four individuals had an interval of three years between moldings.

| Variable | W 279 | W 246 | W 654 | W 288 |
|---|---|---|---|---|
| **Upper second molar** | | | | |
| Total volume lost (mm$^3$) | 9.00 | 3.76 | **109.80** | **9.77** |
| Volume loss per mm$^2$ (mm$^3$) | 0.10 | 0.07 | **0.85** | **0.13** |
| Maximum surface deviation (microns) | 618.50 | 460.10 | **5,984.40** | **1,041.20** |
| **Lower second molar** | | | | |
| Total volume lost (mm$^3$) | **17.58** | **13.33** | 22.81 | 34.06 |
| Volume loss per mm$^2$ (mm$^3$) | **0.20** | **0.19** | 0.26 | 0.41 |
| Maximum surface deviation (microns) | **909.20** | **1,029.30** | 1,358.40 | 1,069.10 |

weak positive trend, with no statistically significant findings across all groupings (Fig. 4; Table 4). Figure 4 also shows the substantial variation among individuals, even within the same interval group and sex. For example, two males (W 731 and W 864), born and died within a month of each other, show substantial differences in tooth wear in the three years

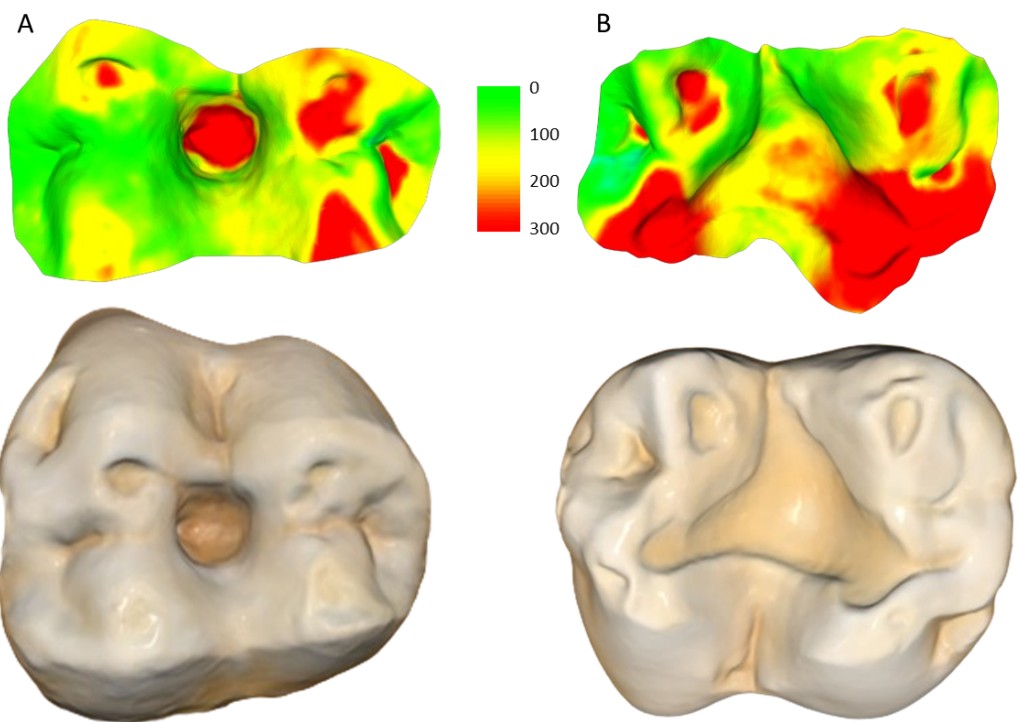

**Figure 3** Specimen W 288: (A) Upper right second molar, showing progression of the carious lesion in the center of the occlusal surface; (B) Opposing lower right second molar, showing unusual wear progression, resembling tooth erosion or severe attrition, although rapid caries progression cannot be ruled out. Scale bar is the distance (microns) lost between the second mold and the osteological specimen (two-year interval). The scale bar is in microns, and represents the distance difference between the molds. Figure created using WearCompare (https://leedsdigitaldentistry.com/wearcompare/).

between molds and at the time of death, suggesting that this trend of dissimilar tooth wear continued throughout life (refer to Fig. 5).

The only specimen that does not exhibit the highest degree of wear on the functional cusps (specifically, the buccal cusps of lower molars and lingual cusps of upper molars), as assessed by the point where the maximum surface deviation occurs, is specimen W 469. In this specimen, a large fracture occurred before the first mold, resulting in a high degree of tissue loss following the fracture. However, tissue loss returns to more moderate levels when a third time point is added using the osteological specimen as the final time point (refer to Fig. 6). In another example, a fracture occurs between the two molds, this time on a functional cusp, and again is associated with severe wear on that cusp (see Fig. 7).

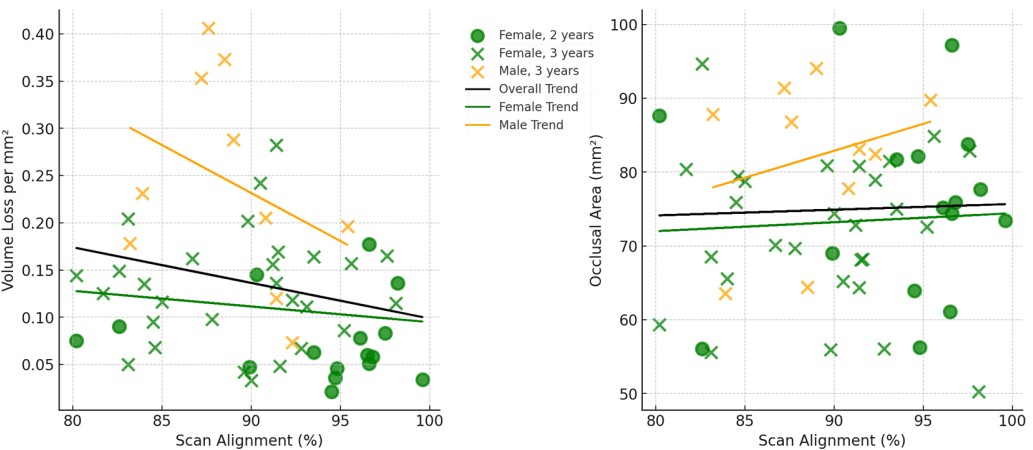

**Figure 4 Relationships between scan alignment and two different dental measurements across all specimens (not including the four individuals with dental caries).** Left: Relationship between scan alignment (%) and volume loss per mm², with regression lines indicating overall and sex-specific trends. Right: Relationship between scan alignment (%) and occlusal area (mm²), with the same corresponding regression lines.

## DISCUSSION

### Technological and methodological considerations

The results of this study provide compelling evidence supporting the applicability of intraoral scanning technology in assessing dental tissue loss in non-human primates. Our results show that the equipment and software development for assessing dental tissue loss in humans can be reliably used in other primate species, in this case a captive baboon sample. Furthermore, our findings indicate an increase in tissue loss with longer time periods, aligning with expectations based on dental wear progression. Contrary to potential concerns regarding the accuracy of these methods in non-contemporary human and non-human primate samples, due to faster levels of tissue loss (*Kaidonis, 2008*; *d'Incau, Couture & Maureille, 2012*), our study suggests that such concerns may be unwarranted.

While efforts to minimize alignment issues between scans, such as those made by the creators of *WearCompare* and similar software, have been largely successful, alignment errors cannot be fully removed due to the absence of fixed reference points in the dentition as teeth wear (*O'Toole et al., 2019b*; *O'Toole et al., 2020*). To mitigate this, *O'Toole et al. (2020)* implemented strict alignment criteria, requiring over 75% of alignment points to be within 25μm of each other. In the present study, the limited wear on the buccal and lingual sides of the teeth meant all alignments were above this threshold. However, as tooth wear progresses over time, achieving precise alignment becomes increasingly challenging. The observed decrease in alignment between the two-year and three-year interval groups, coupled with the decrease in alignment with increase tissue loss (Fig. 4), suggests that the current alignment techniques may surpass the *O'Toole et al. (2020)* defined alignment limits after approximately four to six years of wear, for the techniques and sample used in the present study. However, a larger time series data set is required to find the actual

**Table 4 Summary of the results from linear regression analyses evaluating the relationships between scan alignment (%) and two dental measurements: volume loss per mm² and occlusal area (mm²) of all second molars combined, but not including the four individuals with dental caries.** Each row represents a different subset of data: the overall sample, male specimens, and female specimens.

| Variable | Group | R-Squared ($R^2$) | Correlation (R) | Slop | *P*-value |
|---|---|---|---|---|---|
| **Volume Loss per mm²** | All samples | 0.05 | −0.23 | 0.00 | 0.101 |
| | Male | 0.12 | −0.34 | −0.01 | 0.330 |
| | Female | 0.02 | −0.15 | 0.00 | 0.325 |
| **Occlusal area (mm²)** | All samples | 0.00 | 0.04 | 0.08 | 0.801 |
| | Male | 0.07 | 0.26 | 0.73 | 0.475 |
| | Female | 0.00 | 0.06 | 0.12 | 0.704 |

A

B

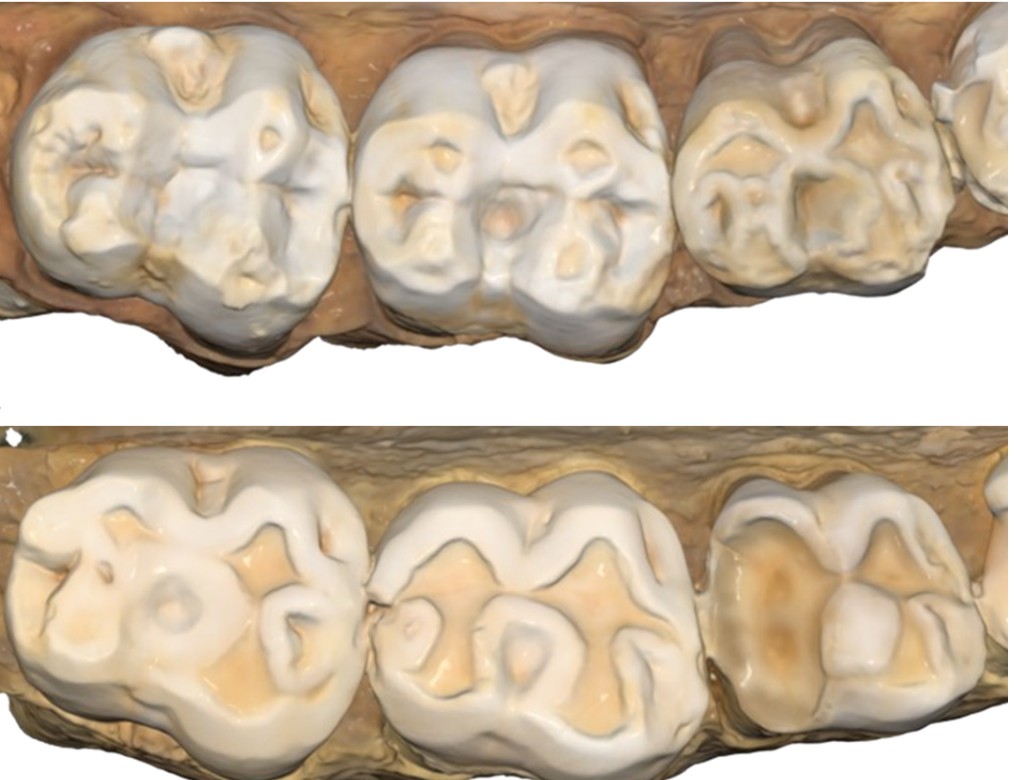

**Figure 5 Wear comparisons of two males.** (A) Upper left molars of specimen W 864; (B) upper left molars of specimen W 731. Both individuals were born in April 1988 and died in January/February 2009. In both cases this is a 2D image taken from 3D models of the osteological specimens (created using the intraoral scanner), showing how specimen W 731 has substantially more wear than W 864, despite being almost identical in age.

limit of these alignment techniques. Additionally, the ability to align scans accurately will not just vary between samples due to difference in tooth wear, but also in relation to the resolution and accuracy of the scans themselves (discussed in more detail below).

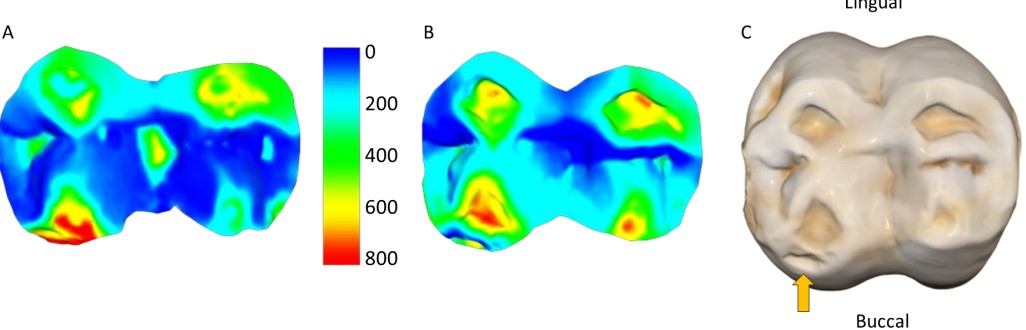

**Figure 6  Upper right second molar (W 469).** (A) From age 10 to 13; (B) from age 13 to 17; (C) tooth at the time of the individual's death at 17 years old (2D image from the 3D model created using the intraoral scanner). Orange arrow highlights the antemortem tooth fracture on the mesial-buccal cusp, which shows substantial tissue loss in (A) but low tissue loss in (B). Scale bar is the distance (microns) lost during the three (A) and four (B) year intervals. Figure created using WearCompare (https://leedsdigitaldentistry.com/wearcompare/).

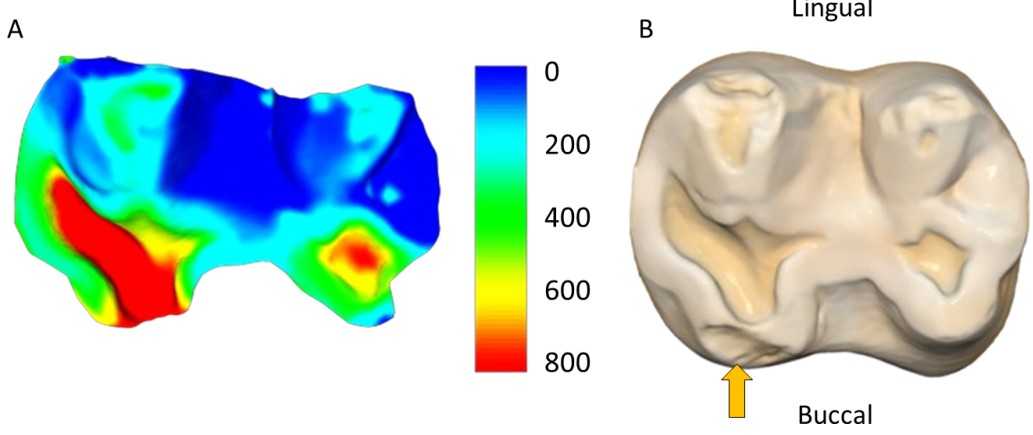

**Figure 7  Specimen W 231 (lower left second molar).** The mesial-buccal cusp displays substantially more tissue loss in the three-year interval between the molds than other cusps (A), in association with a large fracture that formed during this time (orange arrow in (B)). (B) The tooth of the osteological specimen, three months after the second mold was complete. Scale bar is the distance (microns) lost between the two molds. Figure created using WearCompare (https://leedsdigitaldentistry.com/wearcompare/).

It is noteworthy that the alignment of the baboon scans in our study exceeded that of clinical human samples used by *O'Toole et al. (2020)*, as evidenced by fewer individuals being excluded due to alignment issues. Several factors may contribute to this phenomenon. Firstly, although the baboons exhibit substantially more occlusal wear than the human sample, it is possible that erosion, which is likely less prevalent in these baboons compared to attrition or abrasion, contributes to preservation of the buccal and lingual tooth positions over time. Secondly, human molars possess a more bulbous morphology than baboon molars, potentially affecting the ease of aligning their crowns. Thirdly, regular tooth brushing in contemporary human samples, especially if

brushing follows an erosive challenge, may remove sufficient buccal/lingual tissue over a three-year period (*Addy & Hunter, 2003*; *Voronets & Lussi, 2010*; *Bartlett et al., 2013*), making alignment more challenging compared to non-human primates. Lastly, it is important to note that differences in scanners used, and how 3D models are generated (*i.e.,* the software and algorithms used) and the samples themselves (*e.g.,* artefacts or dental movement/pathologies), may make direct comparisons difficult.

Further research with additional samples is warranted to explore these potential hypotheses. However, even without tooth brushing, movement of food by the tongue and cheeks against the teeth is sufficient to cause a small amount of wear on non-occlusal surfaces (*Fox, Juan & Albert, 1996*; *Ungar & Teaford, 1996*), and so in every sample there will be a limit to the interval possible for these techniques. Our study, in conjunction with the findings of *O'Toole et al. (2020)*, suggests that studies aiming to investigate wear progression over extended periods should consider using multiple scans, such as conducting a new scan of the dentition every two to three years. It is also worth noting that it has been possible to assess very low rates of tooth wear in human samples. Consequently, in heavily worn non-human primate samples, shorter intervals between scans may provide valuable insights into short-term wear progress, including seasonal changes in diet or behavior.

Despite the high trueness and precision (and therefore accuracy) of intraoral scanners, ongoing research aims to further enhance scanning technology by addressing potential differences in resulting 3D models based on various factors, including scanner model, color variation, distance from the object, scanning pattern, and ambient temperature (*Flügge et al., 2016*; *Abduo & Elseyoufi, 2018*; *Kihara et al., 2020*; *Amornvit, Rokaya & Sanohkan, 2021*; *Revilla-Leon et al., 2021*; *Revilla-León et al., 2022*; *Afrashtehfar, Alnakeb & Assery, 2022*). This field of technological advancement is rapidly evolving, with each new model, and software update, bringing significant progress. The model used in the present study, the Medit i700, is widely regarded as extremely reliable for complete arch scanning, with trueness comparable to that of high-end desktop scanners (*Borbola et al., 2023*). Variation in trueness and precision has however been found to vary in the literature, including in association with scanning conditions (*Borbola et al., 2023*; *Giuliodori, Rappelli & Aquilanti, 2023*; *Tran et al., 2023*; *Sorrentino et al., 2024*).

In this study, we used the high-resolution (HD mode) setting for all scans. According to the manufacturer, this mode enhances model detail—or resolution—without affecting the specified accuracy (10.9 $\mu$m $\pm$ 0.98). While precision, trueness and accuracy of intraoral scanners are often reported by manufactures and in publications, resolution receives much less attention (*Desoutter et al., 2024*). Indeed, it is typically difficult to get information on the resolution of intraoral scanners from manufacturers (*e.g.,* personal communication, 2024, with Medit Support: "the creation of the mesh and mapping for triangulations, this process is proprietary, protected by our patent, and involves the implementation of the software and the functioning of the AI"). How resolution should be interpreted for 3D surface models also varies in the literature, although intraoral scanners produce models consistent with the range of resolutions of 3D dental models created using other scanning techniques (*i.e.,* 10–100 $\mu$m; *Medina-Sotomayor, Pascual-Moscardó & Camps, 2018*; *Berthaume, Winchester & Kupczik, 2019*; *Desoutter et al., 2024*).

Studies examining how scanning accuracy and resolution, and associated processing procedures, influence estimates of tissue loss are warranted, similar to those performed in other areas of dental and osteological research (*e.g., Profico et al., 2016; Veneziano, Landi & Profico, 2018; Berthaume, Winchester & Kupczik, 2019; Morley & Berthaume, 2023*). Direct comparisons between different techniques are also warranted, including to assess if any consistent variation in tissue loss is found in the same samples using different methods (both variation among intraoral scanners, but also with surface models generated through different techniques), and/or processing techniques. Ultimately, a standardized method for researchers to assess accuracy and resolution individually for each study, such as proposed by *Desoutter et al. (2024)*, will be crucial before detailed comparisons across scanners, software and processing techniques can be made.

This study predominantly uses scans of dental casts. Although both methodologies produce high resolution 3D models (as presently defined), molding teeth followed by casting, followed by scanning with an intraoral scanner, will undoubtedly lead to some (even if slight) reduction of scan quality (*e.g.*, through the inclusion/accumulation of artefacts). Additionally, as discussed above for creating virtual 3D models of teeth, molding and casting comes with its own variation in end product depending on materials used and methods invoked (*Fiorenza, Benazzi & Kullmer, 2009*). But the less-reflective surface of casts may also increase scan detail/quality, even if intraoral scanners can reliably collect data from enamel. Therefore, as observed in clinical contexts over recent years, it is imperative to compare results between research groups using different techniques to confirm any inter-study variation in non-human primate dental tissue loss data. This also includes for different types of samples, including wet (*i.e.,* scans taken on living individuals), dry (*i.e.,* scans of osteological specimens), and casts. The size of the specimen is also something that requires further research, not just with the perspective of tissue loss progression, but also potential scan alignment influences (Fig. 4; see also *Balolia & Massey, 2021*).

Despite these limitations, resulting variation in tissue loss data between methods is likely to be small. For example, differences in intraoral scanner used, alternative methods can yield similar tissue loss results (*e.g., García et al., 2022; Branco et al., 2023; Schlenz et al., 2023*). Indeed, any resulting differences should be expected to be minimal, given the demonstrated high accuracy of intraoral scanners in assessing tissue loss, assessed in relation to Micro-CT scanning (*Mitrirattanakul et al., 2023*). Furthermore, intraoral scanners have proven highly reliable in detecting small amounts of dental tissue loss in a clinical setting (*e.g., García et al., 2022; Marro et al., 2022; Schlenz et al., 2022*).

Other surface scanners, such as the commonly used Artec 3D Space Spider, have been evaluated for their suitability in dental scanning. In two recent studies comparing intraoral scanners with an Artec 3D Space Spider scanner, one suggested the former as better for scanning teeth (*Piedra-Cascón et al., 2021*), whilst the other concluded although both produce similar results, the Artec 3D Space Spider should be considered the "gold standard" for creating 3D models (*Winkler & Gkantidis, 2020*). However, significant negatives of scanners such as the Artec 3D Space Spider have also been pointed out, including its large size and subsequent difficulty in scanning dentitions. Additionally, and most importantly, to produce the "gold standard" results in the *Winkler & Gkantidis*

*(2020)* study, a coating was required to be sprayed over the specimens, and the authors suggest that without such methods accurate readings are not possible. Such a procedure is unlikely to be permitted for many osteological collections. Therefore, the major benefit of an intraoral scanner is that it is designed with dentitions in mind, and therefore collects images reliably without coating teeth. The small size also makes it easy to move around a dentition whether you are scanning a live individual, a cast, or an osteological specimen. Additionally, in many applications of these methods, a living individual will be involved, at least for one of the time points, meaning a larger scanner, such as an Artec 3D Space Spider would not be feasible.

The versatility of intraoral scanners extends beyond dental tissue loss assessment, as highlighted by recent studies in archaeological osteological collections (*Colleter et al., 2023*). These scanners offer non-destructive 2D and 3D imaging capabilities, eliminating the need for direct anthropometric measurements and reducing specimen handling. In addition to being less destructive compared to traditional methods involving calipers or molding materials, the 3D models generated by these scanners can be shared, significantly minimizing the handling of fragile specimens (*Omari et al., 2021*; *Colleter et al., 2023*). Moreover, the scanning process for osteological or cast dentitions typically takes less than five minutes, making it is also vastly quicker than Micro-CT scans for generating surface 3D models. Intraoral scanners also excel at distinguishing color and texture variations, thereby offering potential use for paleopathology studies. Given their diverse capabilities, intraoral scanners can serve as additional or complimentary tools in primatology, osteoarcheology, and biological anthropology research contexts. Their non-destructive nature, speed, and accuracy may make them useful for various applications beyond dental tissue loss assessment.

Deciding on which intraoral scanner to purchase or use can be a difficult process, with scanners costing from under $10,000 to over $50,000. Although intraoral scanners generally offer high precision and accuracy, variations exist among different devices (*e.g.*, *Amornvit, Rokaya & Sanohkan, 2021*). Each new model released by manufacturers tends to improve these metrics, yet it is the additional features that often distinguish one generation from the next. These enhancements can include the transition from corded to cordless designs, improved color accuracy, faster scanning speeds, streamlined calibration processes for color recognition, reduced warm-up times, smaller devices or tips, expanded software applications, and improved resolution of the 3D models generated. Many of these advanced functionalities may not be directly applicable, or important, in osteological or archaeological studies due to specific research requirements. A significant advantage of the scanner utilized in our research is the use of free software that supports the export of scan data in multiple formats. Nevertheless, some commercial scanners come with mandatory subscription fees and software that may not be ideally suited for scanning in a non-clinical setting. The high initial expense of these devices might currently limit their widespread adoption in non-clinical fields, but prices are decreasing. Moreover, the growing use of these devices in clinical and laboratory settings presents opportunities for collaborative research. This approach also allows for the separate purchase of scanner tips, thus mitigating the risk of cross-contamination.

## Tissue loss in captive baboons

The average tissue loss values produced in this study can serve as a foundational reference for future investigations, whether on captive primates or their wild counterparts. A recent study by *O'Toole et al. (2020)*, based on a clinical study of contemporary humans that had high levels of erosive tooth wear, found relatively uniform levels of tooth wear across the dental arcade. The only molar in *O'Toole et al.*'s (*2020*) study with comparable data is the upper first molar, which showed an average total volume of 2.53 mm$^3$ over a mean period of three years. In relative terms, accounting for tooth size, this equates to a mean volume loss per mm$^2$ of surface analyzed of 0.03 mm$^3$. Therefore, in our study, the tissue loss observed in the upper second molars of baboons over a similar three-year interval is typically over four times greater in both absolute and relative terms. The average rate of dental tissue loss through wear is likely even lower in most contemporary humans, considering the participants in *O'Toole et al. (2020)* were considered at risk for high levels of erosive tooth wear. Indeed, although using slightly different techniques, a recent study by *Schmid et al. (2024)* found less tissue loss in individuals that have undergone orthodontic treatment than the baboons in the present study, even after a period of over nine years. These observations align with the prevalent soft food diet typical in most parts of the world today. In contrast, the baboons in our study showed macroscopically visible tissue loss even in the two-year interval group. This higher rate of tooth wear in the SNPRC baboons, compared to the humans in the dentistry studies, may be attributed to the high-fiber content of the monkey chow lab feed. However, differences in the way baboons and humans masticate foods (including jaw movements, forces and occlusion), may also be important to consider and compare in future studies. Additionally, social behaviors and differences in dental morphology may also be crucial factors to consider.

The comparison of tissue loss between carious and non-carious teeth offers insights into the interplay between pathological conditions and wear dynamics in primate populations. The observed reduction in tissue loss in the opposing molar of some individuals with carious lesions suggests the presence of compensatory mechanisms within the masticatory system. However, in one case, severe tissue loss is evident in the opposing molar, higher than in the carious tooth, and indicates potential atypical wear patterns that may be related to the carious lesion in the opposing tooth. This particular case looks very similar to lesions that may be described as erosion (or a combination of erosion and attrition), in a clinical human setting, although a severe form of dental caries cannot be ruled out. Similarly, in the two cases where clear and large antemortem fractures have occurred (one before the first mold and the other between the two molds), the tissue loss across the occlusal surface is substantially different than in other individuals with no fractures. These observations underscore the complex interrelationships between different types of dental tissue loss, such as erosion, attrition, abrasion, fracture, and caries, and the need for further studies to elucidate these dynamics over time.

Despite all baboons in this study sharing the same diet, we observed significant variations in tooth wear and pathologies. This suggests that differences in physiology, morphology, or behavior—or more likely, a complex combination of factors—might explain these patterns. However, even with a uniform diet, the social dynamics and hierarchy within

the group is also important to consider, as such factors may be associated with different amounts of food intake. The way individual baboons handle food can significantly impact tooth wear. This includes their eating duration each time, whether they drop food to the ground (which could introduce dust or grit), and their interaction with water and food (notably, some baboons have been observed softening food by dropping it into water). Some individuals may place non-food items in their mouth and there is also potential for stress-induced tooth grinding.

Physiological and morphological aspects, such as saliva production, the structure and mechanical properties of dental tissues, and genetic factors influencing chewing, will all also have a direct effect on dental tissue loss progression (*Lucas, 2004*; *Buzalaf, AR & Kato, 2012*; *Mulic et al., 2012*; *Hara & Zero, 2014*; *Towle et al., 2023a*; *Towle et al., 2023b*; *Towle et al., 2023c*). Distinguishing between these various factors can be challenging, as illustrated by the extreme difference in tooth wear of two male baboons shown in Fig. 5. Both these individuals show slight asymmetry in tooth wear between the left and right posterior teeth, but this does not explain the observed differences between these individuals. This highlights that significant variation in tooth wear can occur even with a uniform diet, pointing to the importance of considering food processing habits, masticatory differences, and idiosyncratic behaviors in understanding these patterns. A study utilizing a larger sample size that considers these factors, as well as genetic inputs (*e.g.*, subspecies of *P. hamadryas*, or in a pedigree analysis that incorporates tooth wear) may allow some understanding into the main driving forces that explain such wide variation in tooth wear despite a uniform diet.

The prominent tissue loss observed on functional cusps supports their additional susceptibility to wear, although this study focused on moderately worn molars. Assessment of tissue loss progression in later stages of wear, including the examination of different types of dental tissue loss such as fractures, is warranted (*Towle et al., 2021b*; *Towle et al., 2023a*). It is conceivable that the wear on non-functional cusps may intensify over time, consistent with alterations observed in the plane of occlusion over an individual's lifespan. Assessing wear progression will allow this to be directly assessed, not only in wild primates, but also in controlled laboratory settings using wear simulation machines. Such techniques can also be used to look at the association between tissue loss and other variables (*e.g.*, microwear and underlying enamel microstructure; *e.g.*, *Hua et al., 2015*; *Towle et al., 2023b*). Variation in bite force across the dental arcade, and particularly during the eruption of permanent teeth, can influence wear dynamics over time, potentially reducing tooth wear on individual teeth as more tooth contacts are made (*Lucas, 2004*).

Analyses of tooth wear progression across primate populations and samples have predominantly involved evaluating tooth wear at a single time point (*e.g.*, osteological specimens), either by correlating tissue loss with the age of individuals, or by contrasting wear patterns across different teeth within the same dentition (*e.g.*, *Ozaki et al., 2010*; *Elgart, 2010*; *Galbany et al., 2011*; *Morse et al., 2013*; *Galbany et al., 2014*; *Spradley, Glander & Kay, 2016*; *Pampush et al., 2018*; *Ungar et al., 2021*). These approaches essentially construct a time-series analysis by comparing individuals of varying ages, assuming a continuity in wear patterns across each population/sample. Fewer studies have used more direct

longitudinal methodology, tracking wear progression in the same individuals over time, by recording specific wear scores/parameters or the ratio/percentage of dentine exposure, on multiple occasions (*Froehlich, Thorington & Otis, 1981*; *Teaford & Glander, 1996*; *Phillips-Conroy, Bergman & Jolly, 2000*; *Cuozzo et al., 2010*). This study builds on these studies by introducing more comprehensive techniques for evaluating tissue loss through time. These advancements may be particularly beneficial in research comparing tooth wear with other dentition variables, such as recent studies on dental topography and chewing efficiency (*e.g.*, *Venkataraman et al., 2014*; *Pampush et al., 2018*).

The dynamic between dentine and enamel removal once dentine becomes exposed on the occlusal surface can vary substantially and is influenced by factors such as the tooth crown's position, the mastication cycle, and diet (*Lucas, 2004*). Moreover, the specific material being masticated affects tissue loss patterns, as the ability of one object to indent another is determined by their relative hardness. However, volume loss is also influenced by other factors, such as toughness, Young's modulus, and surface characteristics affecting friction (*Kendall, 2001*). Enamel wear is also heavily influenced by the orientation of the enamel prisms relative to the tooth surface, with prisms perpendicular to the contact surface wearing down more rapidly (*Teaford & Walker, 1983*; *Walker, 1984*). External factors such as the presence of dust, silica, and grit significantly impact the patterns of dental tissue loss. All these factors not only affect occlusal wear, but also interproximal wear (*Wood, Abbott, & Graham, 1983*; *Murphy, 1959*). Taken together, these factors and variables highlight the complexity of assessing tooth wear in primates, and the tissue loss techniques described in this study may help tease apart the different components involved.

Laboratory experiments utilizing wear simulations and tissue loss assessment methods can also provide valuable insights into understanding dental tissue loss dynamics from a fossil or archaeological perspective. By investigating how tissue loss varies across the dental arcade over time and in different species or groups, researchers can discern patterns that may have genetic, dietary, or environmental origins (*Tobias, 1980*; *Macho & Berner, 1994*). For instance, if certain patterns of tissue loss are consistently observed in specific species or populations, it may suggest underlying genetic factors influencing masticatory or tooth morphology, or dietary habits shaping wear dynamics. The techniques described here, originally developed for clinical dentistry contexts, offer a promising avenue for investigating the effects of tooth wear on functional morphology in non-human primates and other species (*Ungar & Williamson, 2000*). These methods are not limited to assessing occlusal wear but have also demonstrated accuracy in evaluating other types of wear, such as non-carious cervical lesions (*Denucci et al., 2024*). By conducting research in both laboratory and wild primate settings, scientists can gain a deeper understanding of the development of atypical tooth wear observed in fossil hominin samples, such as "toothpick" grooves (*Hlusko, 2003*).

The ability to reliably quantify dental tissue loss through non-invasive means offers an innovative approach to investigating wear patterns, adaptive responses, and potential ecological shifts within primate populations. These methods allow us to assess individual cusp wear over time and shed light on how underlying tooth morphology influences wear variation among individuals and populations. Research integrating tooth morphology

and cusp size/presence (*Selig, 2024*) presents a promising avenue for studying the genetic underpinnings of tooth wear in primates. In this context, wear could be considered a phenotype in phylogenetic, hybridization and interpopulation studies. For example, previous studies on baboons, such as those by *Hlusko & Mahaney (2003)*, *Hlusko et al. (2004)*, *Ackermann, Rogers & Cheverud (2006)* and *Hlusko, Lease & Mahaney (2006)*, have demonstrated strong associations between cusp size and shape, overall molar size, and the presence of supplemental teeth with genetic factors. By examining a much larger sample of these baboons with uniform diets and analyzing variation across individual tooth crowns for dental tissue loss data, valuable genetic insights could be gleaned.

Moreover, other large cast and osteological collections offer opportunities for similar studies, including investigations into tooth wear differences between wild and captive primates. Scanning additional dentitions of wild and captive individuals may also be feasible, either as part of general health checkups, or alongside other research objectives in a primatological context. While full arch scanning may take several minutes to complete, targeting specific areas, such as mandibular left posterior teeth, can significantly reduce scanning time to just seconds.

## CONCLUSION

The consistent tissue loss patterns captured by intraoral scanners across individuals validate their efficacy as a valuable tool in primate dental research. These data, obtained from captive baboons fed a uniform diet of hard pellets, can serve as benchmarks for future studies. These findings highlight the versatility of intraoral scanning in non-human primate dental research and open avenues for further exploration. This includes investigating wear progression across dental crowns and arcades in both captive and wild primates, exploring correlations between different types of tissue loss, examining the interplay between tissue loss and microwear analysis, and elucidating the genetic underpinnings of tissue loss variation within and between populations.

## ACKNOWLEDGEMENTS

The authors thank Anderson T. Hara, James M. Cheverud, Marina Martínez de Pinillos, Mario Modesto-Mata, and Arthur Thiebaut for helpful discussions during the formation of this manuscript. We thank the Southwest National Primate Research Center for assistance during the collection of the dental molds.

### Funding

This research was funded by the European Research Council within the European Union's Horizon Europe (ERC-2021-ADG, Tied2Teeth, project number 101054659). Raquel Hernando is funded by Juan de la Cierva postdoctoral fellowship (JDC2022-048511-I). The funders had no role in study design, data collection and analysis, decision to publish, or preparation of the manuscript.

## Grant Disclosures

The following grant information was disclosed by the authors:

The European Research Council within the European Union's Horizon Europe (ERC-2021-ADG, Tied2Teeth): 101054659.

Juan de la Cierva postdoctoral fellowship: JDC2022-048511-I.

## Competing Interests

The authors declare there are no competing interests.

## Author Contributions

- Ian Towle conceived and designed the experiments, performed the experiments, analyzed the data, prepared figures and/or tables, authored or reviewed drafts of the article, and approved the final draft.
- Kristin L. Krueger conceived and designed the experiments, authored or reviewed drafts of the article, and approved the final draft.
- Raquel Hernando conceived and designed the experiments, authored or reviewed drafts of the article, and approved the final draft.
- Leslea J. Hlusko conceived and designed the experiments, authored or reviewed drafts of the article, and approved the final draft.

## Data Availability

The complete dataset is available in the Supplemental File.

## Supplemental Information

Supplemental information for this article can be found online at http://dx.doi.org/10.7717/peerj.17614#supplemental-information.

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
