# Peer review of "Assessing tooth wear progression in non-human primates: a longitudinal study using intraoral scanning technology"

_PeerJ, doi:10.7717/peerj.17614_

## Round 0.1 · original submission · Minor Revisions

Dear Dr. Towle,

I have now received the reviews of your paper on dental tissue loss of captive baboons.

As you can see, the reviewers -- in essence-- requested minor changes, largely focused on expanding and clarifying methodological issues. The one reviewer raised the important point of making your data accessible to other researchers; I would strongly encourage you to accept this suggestion in the interest of open science.

Please provide a detailed rebuttal letter to show how you addressed the issues raised by the reviewers.

Sincerely,
Shaw Badenhorst

Reviewer 1 ·

Basic reporting

I thank the editor for inviting me to review this paper. I feel that this paper is a useful contribution to the literature and will likely be cited often as I imagine this scanning modality will become quite popular. Overall, I have only minor suggestions for improving this manuscript. They are outlined below:
Line 1 (Title) - I would suggest changing the 'primates' to 'baboons' as you are only looking at a single species and not across several primate taxa as I first thought when I read the title. Alternatively, if the authors feel strongly about it saying ‘primates’ I might suggest changing it to “non-human primates”
Line 39- This is clarified later in the text, but at this point, I am not certain what is meant by the 'n' here. Is this individual teeth? A single baboon has 32 teeth, so is this basically two individuals? I am just unclear as to what is meant by the n = 62 here. Not to mention this follows the statement "multiple years", so I would think the parenthetical would be the number of years here. Please clarify.
Line 221- Please provide more info about the regression used and the results of that regression. I.e., R2 value, P value, etc.
Line 250- The authors mention “potential concerns regarding the accuracy…”, could the authors cite relevant literature here?
Line 300-314- I am not sure if the authors would be comfortable discussing the costs of these scanning modalities, but it may be worth some discussion about the potential costs of using one scanner versus another. Certainly, the cost is going to be a major limiting factor for other researchers using one of these tools in the future.
Line 332- should read “high levels of erosive tooth wear” not “high levels erosive tooth wear”.
Line 434- remove comma after citation
Figures generally- Would it be possible for the authors to add an additional figure of an entire cast or an entire mandible? It would be nice to see the details of a more complete scan rather than a partial scan. If this is not possible, that is also fine. It would just be nice as a reader to see more of what the scanner can do.
Figure 3- Could the tooth on the right not be showing a highly progressive carious lesion? It looks to me as though the tooth on the left, given enough time, could eventually look like the tooth on the right.
Data availability- I would strongly recommend the authors consider uploading and making their STL files available on MorphoSource. This would be a great way to share the data ( which is important in its own right), but would also let others see what the scans produced by this method might actually look like.

Experimental design

No Comment.

Validity of the findings

No Comment.

Additional comments

No Comment.

·

Basic reporting

This manuscript is written very clearly, and reports on very interesting findings on the dental tissue loss in baboons over multiple years. I very much enjoyed reading it.

Experimental design

1) You should provide some justification as to the taxonomic classification of the baboons of this particular breeding colony belonging to a single species with two subspecies (Papio hamadryas spp.) as opposed to the recognition of several species of Papio (Papio anubis and P. cynocephalus). See lines 114-116.

I understand that the systematics of baboons is still debated but it would be helpful to the reader what your stance is. Also, given the relatively large variation in your results, perhaps one reason could be due to the taxonomic status of these animals.

Please see some references on this debate:

Gilbert, C. C., Frost, S. R., Pugh, K. D., Anderson, M., & Delson, E. (2018). Evolution of the modern baboon (Papio hamadryas): A reassessment of the African Plio-Pleistocene record. Journal of Human Evolution, 122, 38-69.

Kopp, G. H., Sithaldeen, R., Trede, F., Grathwol, F., Roos, C., & Zinner, D. (2023). A comprehensive overview of baboon phylogenetic history. Genes, 14(3), 614.

Sørensen, E. F., Harris, R. A., Zhang, L., Raveendran, M., Kuderna, L. F., Walker, J. A., ... & Rogers, J. (2023). Genome-wide coancestry reveals details of ancient and recent male-driven reticulation in baboons. Science, 380(6648), eabn8153.

2) Please provide technical details on the precision and resolution of the Medit i700 intraoral scanner and the subsequent surface models (line 139). I think this is rather important when assessing the results of the wear comparisons. Also, while the error of this specific scanner is possibly well defined and within a small margin, please comment on any potential error introduced during the physical molding and casting process of the baboon teeth. Could any error perhaps contribute to the observed variance?

Validity of the findings

I concur with the conclusions of this study. With regard to explaining the large variation in the amount of tissue loss among baboon individuals I would suggest considering that access to the monkey chow food was perhaps not equal in all individuals resulting in differnet amounts of food intake. I wonder whether variations in social rank may play a role here. Also, is the food always ingested dry or does it include softened up pellets as well (e.g. due to drinking water)? Lastly, it would be important to know if the individuals belonged to the same subspecies/species or different taxa (see my comment above), and, if the latter, could this also explain some of the variation observed?

Additional comments

Minor comments:

line 161: I found the statement that "the alignment threshold was not reached" a bit confusing as in Table 1 the scanning alignment values are all above 75%. Perhaps you could rephrase to say that the scanning alignment was always greater than the threshold or so.

lines 176-179: I had some difficulties understanding how you got to the average values based on the results provided in Table 1. It would be useful to add that the average values take into account the tooth position (upper and lower M2)

lines 247-249: I agree in general and I somewhat disagree with the statement that the approach presented here can be reliably used in non-human primate species.

line 270. Interesting hypothesis. Is there any published evidence which you could cite here?

line 347. Perhaps the high variance in the results presented might be due to factors not accounted for such as social rank, sex, age. Also, difference in dental morphology (bilophodonty in cercopithecids vs Y5-pattern in hominids

line 371. I think this statement deserves some references.

Table 1: legend. Please add the sample size for each of the three comparisons (females, 2-years; females, 3-years, males, 3-years)

Figure 2, legend. Please add "in occlusal view after "...in lower second molars". Also the unit is missing from the colour scale bar.

Reviewer 3 ·

Basic reporting

This study by Towle et al. assesses dental tissue loss based on casts and osteological specimens of captive baboons through an innovative approach—measurement of surface deviation in scans taken with an intraoral scanner. The study is well framed to show the relevance of not only the topic (tooth wear) in to studies of non-human primate dietary ecology and interpretation of the hominin fossil record, but also why a method that can easily collect longitudinal data from individuals fills much-needed gaps in our understanding of macrowear progression. The writing is professional and the manuscript structure is clear and easy to follow.

In general, I found the manuscript to contain useful and well-supported information that was nicely documented by tables and figures. I think some expansion is warranted in terms of the data provided in the main text (all raw data for the study is provided in the SOM) and have thoughts about some minor changes to the figures, but overall these were strong points of the work. Generally, I do not take issue with the interpretations and stated relevance of the findings as they’re presented here, and my primary areas of critique are limited to aspects of study methodology (specifically the quantification of scan resolution and alignment accuracy).

I consider my proposed revisions to only be minor in nature.

Experimental design

The research presented here is well within the scope of the journal and of interest to biological anthropologists broadly, particularly those who study the primate dentition. The study made clear how the introduction of data from intraoral scanners may address a difficult-to-fill gap in longitudinal analysis of change in primate dentitions. This is commonly skirted by using cross-sectional samples with individuals representing different ages/stages of wear, but this is generally know to require a variety of assumptions, and the current study does an excellent job of illustrating differences in wear between two individuals of nearly identical age who were fed the same diets throughout their lives.

With respect to experimental design, my primary critiques have to do with the known resolution of the scanner and how this potentially impacts analysis of scan alignment. In Lines 139-140, what does the “high-resolution” setting on the intraoral scanner mean for scan resolution? Understandably this approach produces a surface, not a data set of voxels, but what is the degree of accuracy to which surface features are captured? The website for the Medit i700 places the resolution at around 11 microns. Is this what these scans were captured at? Hopefully there are values that can be reported from the scanner and inserted into the text here. If not, you could assess scanning accuracy directly by scanning a phantom of known dimensions (often a sphere of precise, known diameter, or a set of precision calipers pulled some known distance apart). The accurate knowledge and reporting of scan resolution is important, because it impacts understanding of scan alignment, and there might be differences in how subsets of this data behave with respect to scan alignment.

In Lines 160-161, the authors use a pre-established alignment quality definition that is rather hard to interpret in this context. The level of alignment accuracy developed by the O’Toole et al. studies is designed around human teeth at a particular resolution. Can the authors please explain why this is a valid criterion for a nonhuman primate at the resolution achieved in this study (see above)? It would be best if this rationale is clearly spelled out here within the methods.

Finally, within the Discussion (around Line 261 and the associated paragraph), it really looks like this result is driven by the male specimens. I encourage the authors to fit separate regressions to the male and female points and carefully assess the slope confidence intervals (it is fine to include the entire sample regression as well, but the distinct subpopulations could have important interpretive differences). If the female regression has confidence intervals that encompass 0 for the slope, then there is no relationship between volume loss and scan alignment in the female sample. Presumably, the male-only regression will show a much steeper decline than the current full-sample regression. I agree that the authors have chosen a good approach to correct wear for size (volume loss/mm2), but perhaps scan alignment is partially driven by overall size? A regression of scan alignment against surface area (of the scan derived from the first mold) would be a fitting test of this. Needless to say, there should be a distinct table (‘Table 3’) that reports the summary statistics of each of these regressions (including the full-sample regression currently presented).

As a last methodological concern, I wondered if there is any relevant demographic information about study individuals that should be reported in the main text. I later realized that all of the information I was curious about is provided in the SOM, but I think some of the sample demographics warrant reporting in the main text. In particular, we might expect some age dependence in rate of dental tissue loss if the enamel crown becomes perforated, so knowing the ages of first and second sample could be of great interest to readers. Hlusko additionally reported some differences in male and female enamel thicknesses within this sample, and thus the sex would be helpful to know. A new ‘Table 1’ reporting each specimen’s number, sex, caries presence, age at first mold, age at second mold, and age at death would be quite useful – the authors can simply construct it from the relevant columns in the SOM (and leave the SOM table untouched).

Validity of the findings

I found the findings of this study to be generally well supported. I think the authors should further investigate the potential link between scanning alignment and overall specimen size as I described above. Additionally, the language about the beneficial impacts of intraoral scanners to the field seems a bit extreme in Lines 324-327. That being said, the results did not seem overstated or overinterpreted in the discussion and conclusions.

I was surprised that little attention was given to the fundamental differences in how this data was collected (waving an intraoral scanner above dried casts or osteological specimens) vs. how data is typically collected by intraoral scanners. The primary studies providing data quality criteria to this study were based on clinical human dental data, which is collected under very different scenarios! Given the partial goal of this work in advocating for those who study nonhuman primate dentitions to try these techniques, I think it is important to note this distinction. In Line 265 and the associated paragraph of the discussion, I felt it was relevant to bring up how the data of O’Toole et al. would have been collected—primarily in patients’ mouths! Presumably this adds numerous difficulties to the data collection process that are not present when dealing with a cast. I would assume it’s much easier to capture more of the specimen being scanned and that a cast offers much less reflectance than the wet oral environment, improving scanning accuracy. Discussing these differences head-on would bring further the authors’ aim of distinguishing the use of this technology in a domain that is different from what it was designed for (but where it shows great promise).

Additional comments

Below I mention a few minor line comments:

Abstract: An N is given for the teeth in the study, but not for the number of scans. It would be useful here to mention the N for scans (and not leave readers to infer it much later in your methods section).

Line 91: O’Toole et al. 2019 reference missing ‘a’ or ‘b’ – presumably this is meant to be ‘a’?

Line 96: Unclear what is meant by a “well-defined” sample.

Line 107: Really unsure why we’re diving into sociopolitical quagmire of ‘western’ cultures and diets here, and unclear how this is meant to be interpreted or if it is accurate. Do the authors perhaps instead mean ‘people consuming generally soft, heavily-processed diets [which are now commonplace globally]’?

Lines 140-041: Writing unclear. The scans were not made years apart. This sentence should be fixed to read something along the lines of ‘In addition to the two scans of each individual (from the baseline and secondary casts), for three specimens…’

Line 149: Which O’Toole et al. 2019?

Line 156: Which O’Toole et al. 2019?

Line 176: “Both observation groups” is confusing here. The sample has been described as being divided into carious and non-carious individuals, and then divided among non-carious into those with 3-year mold intervals and those with 2-year. Presumably Table 1 is for those in the non-carious category? The text should make this clear. Furthermore, Table 1 presents us with three groupings – not two! Please be more explicit here about what results are presented in Table 1.

Table 1: Is Table 1 only for those specimens in the non-carious category? The table header should make this clear. Including a top row of N in this table would be helpful for interpretation of the results.

---

## Round 0.2 · accepted · Accept

Dear Dr. Towle,

Thank you for submitting the revised manuscript. In my opinion, you have adequately addressed the minor comments and suggestions raised by the reviewers. I am also pleased that the raw data will be made available to other researchers.

Sincerely,
Dr. Shaw Badenhorst